# Occupational injury prevalence and predictors among small-scale sawmill workers in the Sokoban Wood Village, Kumasi, Ghana

Felix Agyemang Opoku[1], Douglas Aninng Opoku[1,2]*, Nana Kwame Ayisi-Boateng[3,4], Joseph Osarfo[5], Alhassan Sulemana[6], Sheneil Agyemang[4], Obed Kwabena Offe Amponsah[7], Michael Tetteh Asiedu[1], Robert Gyebi[8], Peter Agyei-Baffour[9]

1 Department of Occupational and Environmental Health, School of Public Health, Kwame Nkrumah University of Science and Technology, Kumasi, Ghana, 2 Allen Clinic, Family Healthcare Services, Kumasi, Ghana, 3 Department of Medicine, School of Medicine and Dentistry, Kwame Nkrumah University of Science and Technology, Kumasi, Ghana, 4 University Hospital, Kwame Nkrumah University of Science and Technology, Kumasi, Ghana, 5 Department of Community Health, School of Medicine, University of Health and Allied Health Science, Ho, Ghana, 6 Department of Environmental Science, College of Science, Kwame Nkrumah University of Science and Technology, Kumasi, Ghana, 7 Department of Pharmacy Practice, Faculty of Pharmacy and Pharmaceutical Sciences, Kwame Nkrumah University of Science and Technology, Kumasi, Ghana, 8 Department of Epidemiology and Biostatistics, School of Public Health, Kwame Nkrumah University of Science and Technology, Kumasi, Ghana, 9 Department of Health Policy, Management and Economics, School of Public Health, Kwame Nkrumah University of Science and Technology, Kumasi, Ghana

* douglasopokuaninng@gmail.com

## Abstract

### Background

Sawmill workers are at increased risk of occupational injuries due to their exposure to workplace hazards. However, little is known about the burden of occupational injuries among them in Ghana. Understanding its prevalence and associated factors is necessary to design appropriate interventions to improve workers' health and safety. This study sought to determine the prevalence and factors associated with occupational injuries among small-scale sawmill workers at Sokoban Wood Village, Kumasi.

### Methods

A cross-sectional study was conducted among 138 small-scale sawmill workers from December 2020 to January 2021. Data was collected on demographic and work-related characteristics including age, sex, personal protective equipment (PPE), workspace design, and lighting. The primary outcome was the prevalence of occupational injuries in the 12 months preceding the survey. Logistic regression method was used to assess for independent predictors of occupational injuries, and associations were deemed statistically significant at p < 0.05.

### Results

Approximately 66.7% of the workers experienced occupational injuries within the 12 months preceding the study. Cuts (69.6%) were the most commonly reported injuries. Injuries were

**Data Availability Statement:** The dataset supporting the conclusions of this article is included within the article (and its additional file.

**Funding:** The authors received no specific funding for this work.

**Competing interests:** The authors have declared that no competing interest exist.

mainly caused by machine parts/sharp objects (47.8%) and being hit by logs/objects (46.8%). Only 40.7% of the workers reported always using PPE while legs (38.0%) and hands (37.0%) were the most common body parts injured. The worker's monthly income, poor workspace design and poor lighting had increased odds of occupational injuries while an increase in age was associated with a 5% decreased odds of occupational injuries.

## Conclusion

The prevalence of occupational injuries among the sawmill workers at the Sokoban Wood Village was high, and this calls for prioritization of health and safety at the workplace. Essential measures required include improvements in the safety of machine tools, workspace design and lighting.

## Introduction

Sustainable Development Goal Eight seeks to enhance sustainable economic growth and decent work [1]. This places a lot of significance on comprehending and mitigating exposure to hazards at the workplace. Despite this, most workers, particularly, those in the informal sector work under unfavourable conditions such as poor lighting, inadequate working space, and poor ventilation which predispose them to work-related diseases, accidents and injuries [2, 3]. A report by the International Labour Organization showed that over 2.3 million employees suffer work-related illnesses, accidents and injuries every year; claiming over 6000 lives every day globally [4]. It is not only the individual that suffers from work-related injuries but the country as well with an estimated cost of 1.8% to 6.0% of the Gross Domestic Product [5].

The sawmill industry is one of the most hazardous occupations in the world due to the nature of its activities [6, 7]. It requires that the workers use sophisticated equipment or machines which can serve as a source of hazard to them especially when they are improperly used or used without paying critical attention to safety. A growing body of literature from 2007 to 2020 suggests an increased burden of occupational injuries among sawmill workers globally [8–11]. In British Columbia, a study reported a nine-year occupational injury rate for hospitalization of 5.4 per 1000 person-years between 1989 and 1997 [9]. In sub-Saharan Africa, the prevalence of occupational injuries among sawmill workers has been estimated to be between 14.7% and 42.5% in Ethiopia, Nigeria and Kenya [8, 10, 12, 13]. Contact with machine blades, hand tools, lifting of heavy objects and falls, and being hit by falling objects are the predominantly reported mechanisms of occupational injuries in the sawmill industry [10, 13]. The hand is the body part that is mostly affected by occupational injury among sawmill workers [10, 11, 13].

Inadequate safety regulations, lack of concentration, inadequate work space and lack of health and safety training have been reported as some of the factors that predispose sawmill workers to occupational injuries [10, 11, 14]. Work-related stress has also been reported in a study in Ethiopia by Mulugeta *et al.* to be associated with about five times increased risk of occupational injuries among workers in the woodworking enterprises [10]. Poor adherence to the use of personal protective equipment (PPE) also contributes significantly to the occurrence of occupational injuries and accidents [15]. The use of PPE including eye goggles, helmet, ear plugs, hearing protection devices among others protects the worker from coming into direct contact with workplace hazards thereby reducing the risk of occupational injury occurrence [16].

In Ghana, occupational injuries are a major public health issue, with studies reporting a high injury rate among building construction workers [17], small-scale gold miners [18], and road construction workers [19]. Studies assessing the awareness of occupational hazards, policies and safety at the workplace have been conducted among sawmill workers in Ghana [20–26]. Mitchual *et al.* reported that woodworkers' use of PPE was influenced by their awareness of the dangers of health and safety to their job [20]. Another study conducted by Barfi *et al.* also reported that 42.6% of sawmill workers had eye injuries [24]. About 65.6% (145) and 65.3% (115) of occupational injuries recorded by the Kumasi Metropolitan Area Labour Department, Ashanti Region in the years 2001 and 2003 respectively occurred in the wood processing industry [22]. However, studies that assessed occupational injuries and associated factors among sawmill workers in Ghana are scanty.

A review of the literature showed that only one study reported on the factors contributing to occupational injuries among sawmill workers in Accra, Ghana [27]. The study found that sawmill workers with no formal education had about two times increased risk of occupational injuries and that injuries to the hands and legs (56%) were the most commonly reported occupational injuries [27]. This study, however, did not report on the effect the workspace design and lighting could have on the occurrence of occupational injuries among the sawmill workers. This study sought to assess the prevalence and factors associated with occupational injuries, including the possible effects of workspace design and lighting among small-scale sawmill workers at the Sokoban Wood Village, Kumasi, Ghana. This study sought to bridge the literature gap by exploring the possible effect of workspace design, lighting and other factors on the occurrence of occupational injuries among small-scale sawmill workers.

## Methods

### Study design and setting

A quantitative study approach with an analytical cross-sectional design was adopted to collect data from small-scale sawmill workers in the Sokoban Wood Village in Kumasi, Ghana from 20th December 2020 to 30th January 2021. The Sokoban Wood Village was selected due to it being the biggest wood manufacturing site in Kumasi, Ghana with a total of six small-scale sawmill companies. These small-scale sawmill companies have been registered under the Furniture and Wood Workers Association of Ghana. It has about 900 sheds with over 1500 workers including saw millers, wood processors and sellers, carpenters, lumber traders, hardware traders, and food vendors. Sokoban Wood Village serves as one of Ghana's productive industrial sites.

### Study population

The study population was limited to small-scale sawmill workers at the Sokoban Wood Village in Kumasi.

**Inclusion criteria.** Male and female sawmill workers aged 18 to 60 years who have been working with their respective companies for 1 year or more were included in the study.

**Exclusion criteria.** Workers who were not present on the days of the data collection were excluded from the study.

### Sample size estimation and sampling procedure

The Cochrane formula was used to estimate the sample size for this study [28]:

*Sample size*, $SS = \frac{Z^2 pq}{d^2}$, Equation 1[28]

where Z is the reliability coefficient of 1.96 at a 95% confidence interval, p is the proportion of sawmill workers who sustained occupational injuries, q is (1 –p) and 'd' is the degree of accuracy. Assuming a 78.0% prevalence of occupational injuries among sawmill workers from an earlier study [11], and a 5% degree of accuracy, a sample size of 264 was estimated.

$$Sample\ size,\ S = \frac{1.96^2 \times 0.78 \times (1 - 0.78)}{0.05^2} = 264 \qquad \text{(Eq 1)}$$

We adjusted the sample size to 110 for a finite population, N, of 181 (which is the total number of small-scale sawmill workers at the Sokoban Wood Village) using Eq 2. This approach is well documented in the literature [23, 29–31]. The 181 is the total number of all workers from the small-scale sawmill companies. Knowing this was a finite population, we applied the adjustment for it and got the initial sample size.

$$Adjusted\ sample\ size,\ Ass = \frac{S}{1 + \frac{S-1}{N}} \qquad \text{(Eq 2)}$$

where Ass is the new sample size, S is the estimated sample size, 264, and N is the finite population. Adjusting by 25% for non-response, a total of 138 sawmill workers were recruited for the study. We used 25% because initial reconnaissance assessments showed it was prudent to anticipate that interviews could span over 2 days because of the volume of work for the workers. On this basis, there was no guarantee a worker who had commenced the interview the previous day would come to work the next day. To overcome these uncertainties, we used the relatively high 25% adjustment factor.

A total of 138 sawmill workers were selected through a simple random sampling technique from all existing six small-scale sawmill companies at the Sokoban Wood Village. The total number of sawmill workers who were recruited from each company was derived using probability proportional to size. The proportional allocation of the study participants was derived by dividing the number of sawmill workers in each company by the total number of workers (181) in all existing six companies and multiplied by the estimated sample size of 138 (see S1 Table). We compiled all the staff lists in each sawmill company from their respective administration and entered them into balloting. All those who were selected through the balloting were contacted by the research assistants and scheduled appointments for the interviews.

## Data collection instrument and process

Research assistants were trained and engaged to collect study data, employing a face-to-face interview. The training of the research assistants covered questionnaire (see S1 File) administration in both English and the local Twi languages, the consent process, and the data collection instrument to ensure that there was consistency in the data collected. The training was moderated by a biostatistician and a senior research fellow. A self-reported injury which was related to the sawmill activities in the past twelve (12) months preceding the data collection was the dependent variable of interest. The authors developed the questionnaire through a literature review [25, 31–33]. Data on demographic and work-related characteristics including age, sex, duration of employment, relationship status, the provision of PPE, frequency of PPE use, workspace design, lighting, and occupational injuries including mechanism of injuries, types of injuries, body part injured and injury compensation were collected from study participants. Data on the last occupational injury was collected from participants who had sustained more than one injury during the period. The questionnaire was not validated; however, it was pre-tested among fifteen (15) small-scale sawmill workers in a different district which was not

part of the source population to improve its reliability and validity. All the necessary corrections were effected before using it for the final data collection.

## Variables description

**Outcome variable.** The outcome variable of this study was a self-reported injury associated with any sawmilling activities in the twelve (12) months preceding the data collection.

**Predictor variables.** The predictor variables in this study included age, gender, relationship status, level of education, duration of employment, working hours per week, provision of PPE and frequency of PPE, workspace design, lighting, alcohol consumption etc. Workspace design in this study referred to the space (shed) of the sawmill company and how the tools and logs have been arranged. Quality of workspace design was measured by asking the study participants if the tools and logs have been arranged without impeding accessibility (i.e. whether the space is confined or not) with a binary response of "good" or "poor". A poor workspace design in this study meant that the space was confined (not spacious) or the arrangements of logs and work tools have created limited access for the workers while a good workspace design means that the workspace was not confined (spacious) or the workspace allowed easy accessibility. Lighting at the workplace in this study was measured by asking the study participants whether there was sufficient lighting irrespective of the source if they could accomplish tasks freely without having to either strain their eyes for vision or work in an awkward posture, with a binary response of 'good' or 'poor'. 'Good lighting' in this study meant that there was sufficient lighting at the workplace which allowed workers to accomplish tasks freely without having to either strain their eyes for vision or work in an awkward posture while 'poor lighting' in this study meant that there was insufficient lighting at the workplace which either affected workers vision or working posture.

**Data quality control and statistical analysis.** Epi Info version 7.2.4.0 (Centres for Disease Control and Prevention, USA) was used for both data entry and quality checks including cleaning, double and wrong entries. The data was exported to Stata version 16 (StataCorp, College Station, USA) statistical software package for analysis. Both summary and inferential statistics were used to describe the study data. Continuous variables were summarized and presented using means with standard deviations for normally distributed data and medians with interquartile ranges for skewed data. The Student's t-test was used to compare the difference in means of age for workers who sustained occupational injuries and those who did not. Categorical data were summarized and presented using frequencies and percentages. Univariate and multivariate logistic regression analyses were used to determine the factors associated with occupational injuries among the sawmill workers. The outcome variable, occupational injury was binary-coded as '1' for experienced occupational injury and '0' for did not experience occupational injury. Crude odds ratios were estimated at the univariate level to understand the individual associations between each predictor and the outcome (occupational injuries) variables. The variable 'sex' of the study participants was excluded from the univariate analysis because it was virtually homogenous. The multivariable logistic regression analysis included all variables, both statistically significant ($p \leq 0.05$) and non-significant variables in the univariate regression. The non-significant variables including age, work experience, cadre of work, work hours per week, provision of PPE and level of education were included in the model because they have been reported in the literature to have a significant effect on the occurrence of occupational injuries [17, 18, 34, 35]. A backward stepwise regression analysis (at $p = 0.05$) was used to control the non-significant variables that were deemed important. In the final multivariate model, an association was said to be statistically significant at a p-value <0.05. The Variance Inflation Factor (VIF) was used to assess the multi-collinearity of key

independent variables. There was no evidence of multi-collinearity between the key predictor variables from the VIF results (Mean VIF = 1.19, Maximum VIF = 1.36, Minimum VIF = 1.10) (see S2 Table). The fit of the model was assessed using the Hosmer-Lemeshow test and the results did not detect evidence of poor fit (P = 0.190).

**Ethics approval and consent to participate.** The approval to conduct this study was granted by the Committee on Human Research, Publications and Ethics, School of Medicine and Dentistry, Kwame Nkrumah University of Science and Technology, Kumasi, Ghana with reference number CHRPE/AP/485/20. The application for ethical clearance was done after receiving written approval from the Furniture and Wood Workers Association, Sokoban Wood Village, Kumasi, Ghana. All the study participants provided written informed consent before recruitment.

## Results

### Demographic and work-related characteristics of study participants

The mean age of the workers who reported occupational injuries was 35.0 (±12.9) years while the mean age of those who did not experience occupational injuries was 36.8 (±15.5). Workers with no formal education experienced more occupational injuries compared to those who had tertiary education (19.6% vs 6.5%). Workers who worked for 40 hours and above reported more occupational injuries than those who worked for 20 to 40 hours per week (62.0% vs 38.0%). Workers who reported a good workspace design reported fewer occupational injuries compared to those who indicated a poor workspace design (9.8% vs 90.2%). Workers who reported poor lighting at the workplace experienced more occupational injuries than those who indicated good lighting at the workplace (76.1%, vs 23.9%) [Table 1].

### Occupational injuries, health and safety at the workplace among study participants

Approximately 66.7% of the workers reported sustaining occupational injuries in the 12 months preceding the survey. Of those who sustained occupational injuries, about 47.8% indicated the mechanism of occupational injury was being injured by machine/sharp objects. The most commonly reported occupational injuries were cuts (69.6%) and fractures (18.5%). The body parts that were most affected were the legs (38.0%), hands (37.0%) and arms (14.1%). Of those who experienced occupational injuries, over 17.4% were hospitalized [Table 2].

### Factors associated with occupational injuries among study participants

In the univariate analysis, age, workspace design, lighting at the workplace and monthly income were associated with the occurrence of occupational injuries. Factors that were independently associated with occupational injuries after adjusting for both significant and non-significant variables in the multivariate backward stepwise logistic regression were age (Adjusted Odds Ratio (AOR): 0.95, 95%CI: 0.92–0.98), monthly income of 500–999 cedis (85.76 USD to 171.36 USD) (AOR: 4.44, 95%CI: 1.77–11.17), poor lighting at the workplace (AOR: 4.25, 95%CI: 1.68–16.71) and poor workspace design (AOR: 3.85, 95%CI: 1.27–10.76) [Table 3].

## Discussion

The threats to workers' occupational health posed by the informal sector are receiving greater attention at the global level. Empirical evidence suggests that workers in the informal sector are more prone to occupational hazards than in the formal sector where their activities are

**Table 1. Demographic and work-related characteristics of study participants.**

| Variables | Total<br>n = 138 (%) | Injury<br>n = 92 (%) | No injury<br>n = 46 (%) | P–value |
|---|---|---|---|---|
| **Age (years)** | | | | 0.422 |
| Mean age (±SD) | 35.6 (±12.9) | 35.0 (±11.4) | 36.8 (±15.5) | |
| **Sex** | | | | 1.000 |
| Male | 135 (97.8) | 90 (97.8) | 45 (97.8) | |
| Female | 3 (2.2) | 2 (2.2) | 1 (2.2) | |
| **Level of education** | | | | 0.759 |
| No formal education | 24 (17.4) | 18 (19.6) | 6 (13.0) | |
| Basic | 58 (42.0) | 37 (40.2) | 21 (45.7) | |
| Secondary | 46 (33.3) | 31 (33.7) | 15 (32.6) | |
| Tertiary | 10 (7.3) | 6 (6.5) | 4 (8.7) | |
| **Relationship status** | | | | 0.335 |
| Single | 67 (48.6) | 42 (45.7) | 25 (54.4) | |
| Married | 71 (51.4) | 50 (54.3) | 21 (45.6) | |
| **Monthly income (*GHC)** | | | | 0.032 |
| <500 | 60 (43.5) | 33 (35.9) | 27 (58.7) | |
| 500–999 | 69 (50.0) | 53 (57.6) | 16 (34.8) | |
| 1000+ | 9 (6.5) | 6 (6.5) | 3 (6.5) | |
| **Work experience (years)** | | | | 0.100 |
| <5 | 116 (84.1) | 74 (80.4) | 42 (91.3) | |
| 5+ | 22 (15.9) | 18 (19.6) | 4 (8.7) | |
| Median duration (IQR) | 2 (IQR: 2) | | | |
| **Alcohol intake** | | | | 0.925 |
| Never | 76 (55.1) | 50 (54.3) | 26 (56.6) | |
| Occasionally | 45 (32.6) | 31 (33.7) | 14 (30.4) | |
| Every day | 17 (12.3) | 11 (12.0) | 6 (13.0) | |
| **Work hours per week (hours)** | | | | 0.054 |
| 20–40 | 45 (32.6) | 35 (38.0) | 10 (21.7) | |
| 40+ | 93 (67.4) | 57 (62.0) | 36 (78.3) | |
| Mean work hours (±SD) | 45 (±5.0) | 45.1 (±5.5) | 46.3 (±3.8) | |
| **Cadre of staff** | | | | 0.284 |
| Administrative staff | 18 (13.0) | 10 (10.9) | 8 (17.4) | |
| Non-administrative staff | 120 (87.0) | 82 (89.1) | 38 (82.6) | |
| **Supplied PPE** | | | | 0.320 |
| Yes | 86 (62.3) | 60 (65.2) | 26 (56.5) | |
| No | 52 (37.7) | 32 (34.8) | 20 (43.5) | |
| **PPE usage (n = 86)** | | | | <0.001 |
| Always | 35 (40.7) | 16 (26.7) | 19 (73.1) | |
| Sometimes | 51 (59.3) | 44 (73.3) | 7 (26.9) | |
| **Workspace design** | | | | 0.005 |
| Good | 22 (15.9) | 9 (9.8) | 13 (28.3) | |
| Poor | 116 (84.1) | 83 (90.2) | 33 (71.7) | |
| **Lighting at work** | | | | 0.001 |
| Good | 33 (23.9) | 14 (15.2) | 19 (41.3) | |
| Poor | 105 (76.1) | 78 (84.8) | 27 (58.7) | |

IQR: Interquartile range; SD: Standard deviation;

*GHC 5.83: USD 1.00 per exchange rate in Ghana during the study period: GHC: Ghana cedis which is the local currency; Administrative staff include Administrators, and Drivers; Non-administrative staff include Engineers, Labourers, and Operators/Technicians; PPE: Personal protective equipment

**Table 2. Occupational injuries among study participants.**

| Variables | Frequency, N = 138 | Percentage, % |
|---|---|---|
| **Occupational injury** | | |
| Yes | 92 | 66.7 |
| No | 46 | 33.3 |
| **Mechanism of injury (n = 92)** | | |
| Lifting heavy objects | 5 | 5.5 |
| Hit by log/object | 43 | 46.7 |
| Injured by machine part/sharp object | 44 | 47.8 |
| **Type of injury (n = 92)** | | |
| Abrasion | 6 | 6.5 |
| Cuts | 64 | 69.6 |
| Fracture | 17 | 18.5 |
| Sprain | 5 | 5.4 |
| **Body part injured (n = 92)** | | |
| Arm | 13 | 14.1 |
| Foot | 6 | 6.5 |
| Hand | 34 | 37.0 |
| Head | 4 | 4.4 |
| Leg | 35 | 38.0 |
| **Injury treatment (n = 92)** | | |
| Self-care | 44 | 47.8 |
| Medical attention | 48 | 52.2 |
| **Hospitalization (n = 92)** | | |
| Yes | 16 | 17.4 |
| No | 76 | 82.6 |
| **Duration of hospitalization (days) (N = 16)** | | |
| 1 | 13 | 81.1 |
| 3 | 1 | 6.3 |
| 14 | 1 | 6.3 |
| 24 | 1 | 6.3 |
| **Injury compensation (n = 92)** | | |
| Yes | 16 | 17.4 |
| No | 52 | 56.5 |
| Covered medical bills | 24 | 26.1 |

mostly regulated [3, 36–38]. This study reported on the prevalence and factors associated with occupational injuries among small-scale sawmill workers in the Sokoban Wood Village, Kumasi, Ghana. The key health and safety issues that need urgent attention are workspace design, lighting and being cautious with the use of machines at the workplace. The prevalence of occupational injury among the small-scale sawmill workers was high (66.7%) and the occurrence of injuries was predicted by the age of workers, monthly income, workspace design and lighting at the workplace.

The prevalence of occupational injuries (66.7%) observed in the present study underscores earlier reports that the sawmilling industry is hazardous, especially in low and middle-income countries [6, 7, 39]. Concerning occupational injuries, a range of prevalence rates have been reported among sawmill workers. In southwestern Nigeria, a prevalence of 78% of hand injuries has been reported among sawmill workers [11]. However, in Ethiopia, prevalence rates of 14.6% and 41.6% have been reported among wood processing workers [10, 13]. A study of

**Table 3. Factors associated with occupational injuries among study participants.**

| Variable | Crude OR 95%CI | P -value | Adjusted OR 95%CI | P -value |
|---|---|---|---|---|
| Age | 0.99 (0.96–1.02) | 0.419 | 0.95 (0.92–0.98) | 0.004 |
| **Relationship status** | | | | |
| Single | 1.00 | | - | - |
| Married | 1.42 (0.70–2.88) | 0.336 | - | - |
| **Level of education** | | | | |
| No formal education | 1.00 | | - | - |
| Basic | 0.59 (0.20–1.71) | 0.329 | - | - |
| Secondary | 0.69 (0.23–2.09) | 0.511 | - | - |
| Tertiary | 0.50 (0.10–2.40) | 0.386 | - | - |
| **Monthly income (*GHC)** | | | | |
| <500 | 1.00 | | 1.00 | |
| 500–999 | 2.71 (1.27–5.77) | 0.010 | 4.44 (1.77–11.17) | 0.002 |
| 1000+ | 1.64 (0.37–7.16) | 0.513 | 4.01 (0.62–25.87) | 0.144 |
| **Supplied PPE** | | | | |
| Yes | 1.00 | | - | - |
| No | 0.69 (0.34–1.43) | 0.321 | - | - |
| **Work experience (years)** | | | | |
| <5 | 1.00 | | - | - |
| 5+ | 2.55 (0.81–8.05) | 0.109 | - | - |
| **Workspace design** | | | | |
| Good | 1.00 | | 1.00 | |
| Poor | 3.63 (1.42–9.31) | 0.007 | 3.85 (1.27–10.76) | 0.017 |
| **Lighting at work** | | | | |
| Good | 1.00 | | 1.00 | |
| Poor | 3.92 (1.73–8.88) | 0.001 | 4.25 (1.68–16.71) | <0.001 |
| **Cadre of staff** | | | | |
| Administrative Staff | 1.00 | | - | - |
| Non-administrative staff | 1.73 (0.63–4.72) | 0.288 | - | - |

OR: Odds ratio; CI: Confidence interval; 1.00: Reference category

sawmilling industries in Kenya found a prevalence of 45.1% of occupational injuries/accidents among workers [8]. The difference in prevalence rates in these studies can be attributed to the variations in injury definition and types, recall periods of injuries and differences in exposures in the workplace environment. For instance, the present study assessed all types of occupational injuries which were sustained while the Nigerian study assessed only hand injuries [11] which could explain the higher injury rates in the current study. The present study observed injuries by machine parts/sharp objects and being hit by logs/objects as the commonly reported mechanism of occupational injuries among the small-scale sawmill workers. In Ethiopia, Mulugeta *et al.* also reported contact with machine blades and splintering objects as the commonest mechanism of injury [10].

The predominant type of injuries (cuts and fractures) observed among the sawmill workers in our study is comparable to a study conducted among sawmill workers in Ethiopia [13]. Specifically, the body parts that were mostly affected by the injuries were the legs, hands and arms. The body parts affected by injuries in the present study are similar to what has been reported among sawmill workers in Ethiopia and Kenya [8, 10, 13]. We recommend that the management of the sawmill companies admonish the workers to be cautious by observing all protocols

associated with their usage. Also, management can institute an award scheme for those who went through the year without injuries.

It was observed in the present study that an increase in age was associated with a 5% (AOR: 0.95, 95%CI: 0.92–0.98) decreased odds of occupational injuries among the small-scale sawmill workers. This implies that younger workers were more susceptible to occupational injuries than older workers. This is in line with what has been documented in the literature [40, 41]. Younger workers may not have gathered enough experience and skills to operate workplace machines which could predispose them to occupational injuries. Again, unlike the older workers who may have the ability to detect workplace hazards and raise concerns regarding their health and safety at work, the younger workers may lack that experience which can increase their risk of occupational injuries. This study recommends that management should conduct health and safety training for all young sawmill workers as this could be important for reducing their risk of occupational injuries [31]. The workers could also benefit from increased awareness and education on health and safety and this should be championed by the management.

We found that the monthly income of the workers was associated with occupational injuries which resonates with earlier studies [17, 42]. In the current study, sawmill workers who were on a monthly income of 500–999 cedis (85.76 USD to 171.36 USD) had about four (AOR: 4.44, 95%CI: 1.77–11.17) times increased odds of occupational injuries compared to those who earned less than 500 (85.76 USD) cedis. A possible reason could be that those who earned between 500 to 999 cedis (85.76 USD to 171.36 USD) were engaged in rigorous activities which involved the use of sharp machines which predisposed them to occupational injuries while those who were earning less than 500 cedis (85.76 USD)] may be involved in less demanding activities such as administrative works which could reduce their risk of occupational injuries.

We also observed in the present study that, sawmill workers who perceived their workspace design to be poor had about four times (AOR: 3.85, 95%CI: 1.27–10.76) increased odds of occupational injuries compared to those who perceived it to be good. The employees' health and safety can negatively be impacted by the design of the workspace. One of the key principles of ergonomics is to "fit the job to the worker, not the worker to the job" [43]. The workspace should be designed to match the anthropometric dimensions of the worker. A good workspace design should have adequate space which can enhance free movement or accessibility and stretching of body parts. Other studies have reported that a poorly designed workspace can predispose workers to adverse health outcomes such as occupational illnesses, injuries and accidents [44–46].

In the present study, over three-quarters (76.1%) of the sawmill workers reported that the lighting at their workplace was poor. This is particularly disturbing looking at its deleterious effect on employees' health and safety as this can impede their vision and predispose them to occupational injuries. For instance, it was observed in the present study that workers who reported poor lighting at their workplace had about four times (AOR: 4.25, 95%CI: 1.68–16.71) increased odds of occupational injuries compared to those who had good lighting. Poor lighting at the workplace contributes to occupational injuries such as sprains, fractures and cuts [47]. It is recommended that the Ghana Labour Commission come up with a standard workplace lighting for sawmill companies and routinely conduct monitoring and evaluation to ensure that the companies abide strictly by it.

The current study observed that the use of PPE by sawmill workers was problematic with less than half (40.7%) of those who were supplied PPE at the workplace always using it. The use of PPE at the workplace is one of the most effective ways to prevent the occurrence of occupational injuries and accidents [13, 48], especially in situations where both engineering and

administrative controls are not feasible. Failure to use PPE can expose workers to hazards which can predispose them to occupational injuries [13]. A possible reason could be that those who did not always use PPE at the workplace were very confident in themselves regarding their ability to perform the task without being injured or they were not comfortable wearing them. Another reason could also be that they were not aware or knowledgeable about PPE usage. This study therefore recommends that the management of the sawmill companies should provide the appropriate PPE for all workers and also train them on their use as well as ensure that they use them at all times in the workplace. Management can also institute punitive measures for workers who do not use PPE at the workplace.

## Strengths and limitations of the study

This study provides useful information to inform policy direction that would enhance the health and safety of sawmill workers in Ghana and beyond. Again, this study reports on the influence of workspace design and lighting at the workplace on the occurrence of occupational injuries which had not been previously reported in the Ghanaian context.

The study has some limitations including self-reporting of occupational injuries (not clinically confirmed) which can result in recall bias or misclassification of the outcome. However, to avoid misclassification of occupational injury, study participants were asked to detail how and where the injury occurred. The assessment of occupational injuries in the last twelve months preceding the data collection can result in an underestimation of the prevalence rate due to recall bias (participants had to recall the event in the last 12 months). Some of the confidence intervals around the odds ratio estimates were wide due to the relatively small sample size, hence, we recommend that these results be interpreted cautiously. Again, this study did not assess the types of PPE which were provided to the workers and also did not review existing documentation on health and safety procedures. Also, workspace design and lighting were subjectively assessed. Even though the questionnaire was pretested and fine-tuned following that, the lack of a comprehensive evaluation of validity and reliability is a limitation and the findings ought to be taken with caution. Nevertheless, the findings of this study are insightful for further more comprehensive investigations in the future.

## Conclusion

Our study revealed that the prevalence of occupational injuries among sawmill workers in the Sokoban Wood Village, Kumasi was high which raises concerns for public health. Worker's age, monthly income, workspace design and lighting at the workplace were the determining factors for the occurrence of occupational injuries. The commonly reported mechanism of occupational injuries was injury by machine parts/sharp objects and being hit by logs/objects. We recommend that the management of the sawmill companies should ensure that the arrangements of logs and working tools at the sawmills do not affect accessibility and that lighting at the workplace provides adequate visibility which will not result in workers straining their eyes or working in an awkward posture. It is also recommended that the management of the sawmill companies should ensure regular maintenance of work machines and enforce increased use of PPE by the workers. Future studies can evaluate the impact of regulatory or legislative agencies on the employees' health and safety in the small-scale sawmill industry. This study provides useful information to inform policies to champion employees' health and safety in small-scale sawmill companies.

## Supporting information

**S1 Table. Sample size for small-scale sawmill companies.**
(DOCX)

**S2 Table. Multi-collinearity test results.**
(DOCX)

**S1 File. Questionnaire for study data.**
(DOCX)

**S1 Dataset.**
(XLS)

## Acknowledgments

We acknowledge the support of the Furniture and Wood Workers Association, Sokoban Wood Village, Kumasi, Ghana for conducting this study. We are also grateful to the research assistants for their hard work and commitment towards this study.

## Author Contributions

**Conceptualization:** Felix Agyemang Opoku, Douglas Aninng Opoku, Nana Kwame Ayisi-Boateng, Peter Agyei-Baffour.

**Data curation:** Felix Agyemang Opoku, Douglas Aninng Opoku, Nana Kwame Ayisi-Boateng, Joseph Osarfo, Alhassan Sulemana, Sheneil Agyemang, Michael Tetteh Asiedu, Robert Gyebi.

**Formal analysis:** Douglas Aninng Opoku.

**Methodology:** Felix Agyemang Opoku, Douglas Aninng Opoku, Nana Kwame Ayisi-Boateng, Joseph Osarfo, Alhassan Sulemana, Sheneil Agyemang, Obed Kwabena Offe Amponsah, Peter Agyei-Baffour.

**Project administration:** Felix Agyemang Opoku, Douglas Aninng Opoku, Nana Kwame Ayisi-Boateng, Joseph Osarfo, Alhassan Sulemana, Obed Kwabena Offe Amponsah, Robert Gyebi.

**Supervision:** Peter Agyei-Baffour.

**Validation:** Douglas Aninng Opoku, Nana Kwame Ayisi-Boateng, Joseph Osarfo, Alhassan Sulemana, Michael Tetteh Asiedu, Robert Gyebi.

**Writing – original draft:** Felix Agyemang Opoku, Douglas Aninng Opoku, Sheneil Agyemang, Obed Kwabena Offe Amponsah.

**Writing – review & editing:** Felix Agyemang Opoku, Douglas Aninng Opoku, Nana Kwame Ayisi-Boateng, Joseph Osarfo, Alhassan Sulemana, Sheneil Agyemang, Obed Kwabena Offe Amponsah, Michael Tetteh Asiedu, Robert Gyebi, Peter Agyei-Baffour.

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
