## [Decision Letter · Decision Letter 0]

10 Oct 2023

PONE-D-23-20985Assessment of occupational injury rate and risk factors among sawmill workers in the Sokoban Wood Village of GhanaPLOS ONE

Dear Dr. Opoku,

Thank you for submitting your manuscript to PLOS ONE. After careful consideration, we feel that it has merit but does not fully meet PLOS ONE’s publication criteria as it currently stands. Therefore, we invite you to submit a revised version of the manuscript that addresses the points raised during the review process.

Our reviewers have done their job by giving authors constructive review comments/questions/queries to be addressed including others as articulated and shared below. Some of the critical issues bother on methodology - design, sampling, analysis, generalization issues, scope of study etc. Authors will have to address all the issues raised and every bit and pieces of the reviewers' comments and where there is the need to acknowledge certain limitations in the work, it must be done so with relevant scientific justifications and proofs. 

We look forward to receiving your revised manuscript.

Kind regards,

Bismark Dwumfour-Asare, MSc, PhD

Academic Editor

PLOS ONE

2. We are unable to open your Supporting Information file [ANALYSIS_SAWMILL.dta]. Please kindly revise as necessary and re-upload.

Additional Editor Comments:

We needed to wait on our reviewers, at least four (4) this time round before we could make a final decision at this point in time. The paper, we believe, has received a thorough review as expected and the final outcome is to give authors the opportunity to undertake serious revision before resubmission for consideration. At least all the reviewers have constructive comments for improvement of the paper but most importantly, the critical comments from three (3) reviewers two of whom rejected and one request for a major review.

We hope authors will accept these constructive reviews as part of the rigorous academic quality assurance of any credible journal like PLOS ONE to improve their paper to the acceptable standard demanded by publishing house.

Reviewers' comments:

Reviewer's Responses to Questions

**Comments to the Author**

1. Is the manuscript technically sound, and do the data support the conclusions?

Reviewer #1: Yes

Reviewer #2: No

Reviewer #3: No

Reviewer #4: Partly

2. Has the statistical analysis been performed appropriately and rigorously? 

Reviewer #1: Yes

Reviewer #2: No

Reviewer #3: No

Reviewer #4: No

3. Have the authors made all data underlying the findings in their manuscript fully available?

Reviewer #1: Yes

Reviewer #2: Yes

Reviewer #3: No

Reviewer #4: No

4. Is the manuscript presented in an intelligible fashion and written in standard English?

Reviewer #1: Yes

Reviewer #2: Yes

Reviewer #3: No

Reviewer #4: Yes

5. Review Comments to the Author

Reviewer #1: Reviewer’s comments:

Comment 1. The topic seems practical but some punctuation principles have not been used correctly. Standard punctuation principles should be used throughout the manuscript. It is recommended that the article be edited by a native speaker to improve the article.

Comment 2. It seems that the first sentence of “Results of the Abstract section ” stating “The mean age of the workers was 35.6 (±12.9) years with a range of 18 to 60 years.” is inappropriately placed. It would be better to use it in the “Methods” section whether in the Abstract or in the body manuscript.

Comment 3. What does the AOR mean? It is not recommended to use the abbreviations in the Abstract section (AOR and PPE), also, the original name of the word should be used.

Comment 4. How did you validate your study questionnaire? Did you carry out a validity and reliability analysis?

Comment 5. It seems that the heading of “Characteristics of study participants” would be transferred to the Methods section, as mentioned above. The Characteristics of study participants was are not related to the study results/findings; in fact, they are just descriptive data.

Comment 6. Is it possible to categorize the injuries into very severe, severe, mild, other categories?

Comment 7. What is the difference between the Cude OR and Adjusted OR (AOR)? Please clarify the difference in the manuscript.

Comment 8. Please check the injury rate between males and females (Male 90 (66.7) and female 2 (66.7)).

Comment 9. It is recommended that strategies or solutions which were presented at the end of each paragraph in the “Discussion” section, should be further clarified or explained in more detail, for instance, the acceptable level of lighting, workspace, or even other critical facilities influencing the occupational injuries.

Comment 10. It is suggested that the study questionnaire be made available to the readers as a supplementary material.

Reviewer #2: Assessment Results

The study sought to assess the occupational injury rate and risk factors among workers among wood village in Ghana.

1. The abstract was poorly formatted.

2. Results on the abstract was very scanty and a bit vague

3. Conclusion from the abstract was obvious based on anecdotal evidence and so nothing new was added

4. The objective of the study was not smart enough. Please recast the objective to make it smart.

5. The authors claim they interviewed saw mill companies at Sokoban. However, a cursory look of Sokoban wood village can only reveal small holder saw millers and wood traders. This erroneous impression must be corrected.

6. The authors failed to give an account of the number of small holder saw millers traders they engaged with. Please state the number of registered saw miller shops visited

7. Although the authors employed Cochrane’s formular in estimating the sample size, the sample size looked a bit on the low side, judging from nature of conclusion that the study intends to draw. The sample size used look inadequate.

8. The authors did not state the type of cross-sectional study employed. But judging from the results presented, it seems to me the authors employed quantitative data collection approach. Generating frequency results from such a work cast doubts on the validity of the conclusion. It is therefore suggested that, the authors MUST collect data from another wood production village based on different criteria to server as control.

9. For the quantitative study, the questions asked were a bit scanty. This makes the result a bit small to a made a very valid conclusion. It is suggested that more questions be added to expand the scope of questions.

10. It is also being suggested that some qualitative data should be taken from the study areas to add some contextual complexities to the quantitative results to be presented.

11. The authors claim they conducted multivariate analysis on some proportions of the respondents but the number is missing from the methodology. Please state the number of study subjects you conducted the multivariate analysis on.

12. The authors employed only frequency assessment for Table 1 and 2. This is academically unacceptable for the kind of conclusions you intend to draw. Once a control group will be engaged, at least a Student T test analysis should be carried out to establish the appropriate level of significance in addition to the multivariate analysis.

13. The authors claim they conducted multi variate analysis on both significant and non-significant results. Can you please explain the rationale behind this approach?

14. The authors must clearly state both inclusion and exclusion criteria during data collection

15. The study sought to assess the occupational injury rate and risk factors among workers among wood village in Ghana, however, data was collected from only a small area in Kumasi. This clearly cast doubt on the validity of your conclusion

16. On the basis of the above, I therefore reject the manuscript.

Reviewer #3: 1. In the conclusion part of the abstract section, the presented conclusion is more general and can be presented without conducting research. The conclusion should be derived from the research results.

2. The present study lacks any particular innovation.

3. In Table 1, the number of married people is 41 and single people are 67, so their total is different from the total number of people under investigation. The sum of the mentioned percentages is also not correct.

4. Due to the existence of different types of PPE, it is not correct to mention the general title of PPE in this study. It should be specified what type of PPE is intended by the authors. Which PPE is provided by the employer and which is not. How much is the use of each PPE?

5. According to the results, about half of the workers have declared that safety and health instructions are available, and the other half have declared that the instructions are not available. Determining the level of access to procedures requires reviewing existing documents and cannot be achieved through questionnaires or interviews.

6. According to the results, 89.9% have declared that the working environment is safe, while only 15.9% of the workers have declared that the design of the working environment is good. which indicates a contradiction in the results.

7. Work environment design is a general term. It should be clear what the authors mean by the design of the work environment. What aspects of design are considered?

8. The numbers and percentages presented in Table 3 do not match each other.

9.Contrary to the title, the main and technical risk factors have not been identified in this research.

Reviewer #4: Summary of the research and your overall impression

The research assesses the prevalence and risk factors associated with occupational injuries among sawmill workers in Ghana, specifically at one of the largest woodwork sites in the country, Sokoban Wood Village. The study aims to understand the extent of the problem and identify factors that contribute to these injuries.

The study concludes that:

1. Occupational injuries are a significant issue among sawmill workers in Ghana, with approximately two-thirds of the workers experiencing such injuries.

2. The most common mechanisms of occupational injuries in this population are being injured by machines or sharp objects and being hit by logs or objects.

3. Several risk factors are associated with occupational injuries, including the non-use of personal protective equipment (PPE), poor workspace design, and inadequate lighting.

4. The study found that increasing age and having a secondary education level are associated with a reduced likelihood of experiencing occupational injuries.

The conclusions drawn from this study are that the prevalence of occupational injuries among sawmill workers in Ghana is high, and interventions aimed at improving the health and safety of these workers should focus on younger employees and those with no formal education. Additionally, efforts should be made to enhance workspace design and lighting conditions in the workplace.

Strengths of the manuscript:

1. The study addresses an important occupational health issue in Ghana, shedding light on the prevalence and risk factors of occupational injuries among sawmill workers.

2. It uses a cross-sectional study design with a random sampling approach, which can provide valuable insights into the studied population.

3. The use of multivariate logistic regression allows for the identification of independent predictors of occupational injuries, contributing to a more comprehensive understanding of the problem.

Weaknesses of the manuscript:

1. The sample size of 138 sawmill workers may be relatively small, and the findings may not be fully representative of all sawmill workers in Ghana.

2. The study relies on self-reported data, which may be subject to recall bias and may not capture all instances of occupational injuries accurately.

3. The manuscript lacks information on the specific interventions or recommendations for improving workplace safety beyond general suggestions.

Overall recommendation: The manuscript provides valuable insights into the prevalence and risk factors of occupational injuries among sawmill workers in Ghana. However, it would be beneficial to address the weaknesses mentioned above and provide more specific recommendations for interventions to improve workplace safety. Given the importance of the topic, I recommend accepting the manuscript with major revisions to address these concerns and enhance the overall impact of the study.

2. Evidence and examples

Major Issues

1. Lack of Clear Differentiation from Existing Studies - The introduction does not sufficiently highlight how this study distinguishes itself from previous research, particularly the study titled "Characterizing occupational injuries among sawmill workers of Agbogbloshie timber market, Accra." While it mentions limited studies on occupational injuries among sawmill workers, it fails to specify what these studies did not address or what gaps they left unexplored. This omission makes it challenging for readers to understand the novelty or unique contribution of this study in relation to existing research. The authors should therefore highlight the novelty of the research.

2. Unclear Significance of the Study - The introduction briefly mentions the prevalence of occupational injuries among sawmill workers in Ghana and the studies conducted among woodworking industry workers in Kumasi. However, it does not clearly establish why it is essential to address this issue or why it matters. The authors should elaborate on the significance of this research by explaining the potential impact of the findings on the health and safety of sawmill workers, workplace policies, and public health in Ghana. Providing a clearer rationale for the study would help readers better appreciate its importance.

3. The authors should clearly explain why they chose to use the Cochran's sample size formula instead of the Yamane (1967) formula: n=N/(1+Ne2). Cochran's sample size formula is for large-size populations and not usually used for the population of 181 people as indicated by the authors. Using the above formula would have resulted in a sample size of 125 instead of 110 and with a 25% non-response rate, the sample size will be 156 instead of 138. Clearly, the approach used in this study has reduced the sample size which affects the generalizability of the findings of the study. This should be stated as part of the limitations of the research.

4. Backward Stepwise Approach with a p-value of 0.1: Using a p-value of 0.1 for variable selection in backward stepwise regression is less common and less stringent than the conventional p-value of 0.05. This approach may increase the risk of including variables that are not truly significant, potentially leading to overfitting and less robust results. It's advisable to use a more standard significance level of 0.05 for variable selection to ensure the reliability of the model. The authors should justify the selection of the p-value of 0.1 for variable selection.

5. Including Non-Significant Variables: The methodology mentions including both significant and non-significant variables from univariate analysis in the multivariable logistic regression. While it can be valuable to include theoretically important variables regardless of their statistical significance, it's crucial to justify why these non-significant variables are included. The rationale should be provided based on prior literature or theoretical considerations instead of only listing these sources.

6. Clarify the Analysis Process: In the methodology section, the authors should explicitly clarify the sequence of analyses and their purposes. They should start by explaining that crude odds ratios were calculated at the univariate level to understand the individual associations between each independent variable and the outcome (occupational injuries). Then, describe the backward stepwise approach for variable selection in the multivariate model.

7. Specify the Purpose of Crude Odds Ratios: The authors should clearly state that the purpose for the calculation of crude odds ratios at the univariate level

8. Multivariate Model Development: The authors should describe the development of the multivariate logistic regression model using the backward stepwise approach.

9. Handling Categorical Data: The authors should explain how categorical data were coded for inclusion in the model and provide details on any reclassifications or transformations performed on the categorical variables. They should also clarify the reference categories for each categorical variable used in the analysis.

10. Handling Missing Data: The authors should describe how missing data were handled in the analysis. Specify whether any imputation methods were used, or if data were excluded, provide reasons for data exclusions. Transparency in handling missing data is crucial for the validity of the results.

11. Testing for Collinearity: The authors should explain how they tested for multicollinearity among the independent variables prior to fitting the multivariate model. They should describe the methods used to assess collinearity (e.g., variance inflation factor or correlation matrices) and report the results. If collinearity was detected, they should explain how it was addressed.

12. Presentation of p-Values: The authors should present p-values for each variable in the final multivariate model. This is essential for assessing the statistical significance of each predictor in relation to the outcome.

13. The conclusion of this research is a repetitive summary of the findings already presented. To improve on this, the following are suggested for the authors:

• Restate the objective and indicate whether it was answered or achieved.

• Discuss the implications of the findings and explain why they matter, including any practical applications or theoretical implications.

• Acknowledge any limitations or weaknesses of the study and suggest directions for future research, including any areas where further investigation is needed.

Minor issues

Missing references

1. The last section of this part of the manuscript should be supported by a reference or removed “employment to the majority of the youth in Kumasi, Ghana”. The assertion that the Sokoban wood village provides employment to the majority of youth in Kumasi should be verified since it could be misleading.

Technical clarifications

1. The authors should define the roles of the sawmill workers involved in the study. For instance, what do the authors mean by Engineers, Operators and Technicians? This is not clear and could be used to refer to the same type of workers.

2. The authors should provide a justification for excluding those who had worked for less than a year? Due to the inexperience of new hires, they are exposed to more work injuries compared to experienced ones. This has been well established in existing literature.

3. How was the study population obtained? The authors should indicate the source of that information.

4. The authors should indicate how many sawmill companies were available at the study site and how they selected the six sawmill companies for the study. Do the six companies represent only, say, 10% of the total number of companies at Sokoban? This should be made clear in the manuscript so the applicability of the findings from the research can be identified by readers.

5. How was the probability proportion approach used? How many respondents were selected from each company, and how many were there in total?

6. Table 2: The variables used for each of these parameters should be explained. Workplace environment (Safe, Not safe) What do the authors mean by safe workplace environment and vice versa?

7. Workspace design (Good or poor) What do the authors mean by good workspace design and vice versa?

8. Lighting at work (Good or poor) What do the authors consider as good lighting at work and vice versa?

Data presentation

1. The authors should provide the questionnaire used for data collection as a supplementary file to the manuscript.

2. Regarding the assessment of the availability of health and safety procedures, it may be more efficient to determine whether these procedures exist at the company level rather than asking each individual worker. It could be more informative if the authors report the presence or absence of these procedures for each of the six companies separately. This would clarify how many out of the six companies have established health and safety procedures and how many do not. Alternatively, if this information cannot be obtained accurately, it might be advisable to consider excluding this aspect from the results to prevent potential confusion in its current format.

6. PLOS authors have the option to publish the peer review history of their article (what does this mean?). If published, this will include your full peer review and any attached files.

Reviewer #1: No

Reviewer #2: No

Reviewer #3: No

Reviewer #4: No

---

## [Author Response · Author response to Decision Letter 0]

10 Dec 2023

Reviewer #1: Reviewer’s comments:

Reviewer’s Comment: The topic seems practical but some punctuation principles have not been used correctly. Standard punctuation principles should be used throughout the manuscript. It is recommended that the article be edited by a native speaker to improve the article.

Authors Response: This has been done as suggested. Thank you. 

Reviewer’s Comment: It seems that the first sentence of “Results of the Abstract section” stating “The mean age of the workers was 35.6 (±12.9) years with a range of 18 to 60 years.” is inappropriately placed. It would be better to use it in the “Methods” section whether in the Abstract or in the body manuscript.

Authors Response: The sentence in question has been taken off.

Reviewer’s Comment: What does the AOR mean? It is not recommended to use the abbreviations in the Abstract section (AOR and PPE), also, the original name of the word should be used.

Authors Response: The results section in the abstract has been revised to reflect these changes.

Reviewer’s Comment: How did you validate your study questionnaire? Did you carry out a validity and reliability analysis?

Authors Response: The questionnaire was not validated, however, the study adopted rigorous procedures such as the involvement of highly experienced professionals including an epidemiologist, occupational health and safety officer, biostatistician and a public health practitioner to review its content. This was done to improve its validity and reliability. Again, the questionnaire was pre-tested among fifteen (15) sawmill workers in a different district that was not part of the source population to improve on the internal validity and reliability before using it for the data collection. 

Reviewer’s Comment: It seems that the heading of “Characteristics of study participants” would be transferred to the Methods section, as mentioned above. The Characteristics of study participants was are not related to the study results/findings; in fact, they are just descriptive data.

Authors Response: We are grateful to the reviewer for these comments. However, we respectfully would prefer to maintain the “characteristics of the study participants” under the results subheading as this is the convention. As you are well-aware the methods section describes how a study was conducted to enhance reproducibility. 

Reviewer’s Comment: Is it possible to categorize the injuries into very severe, severe, mild, other categories?

Authors Response: This will be difficult to do as we did not collect the full complement of data needed to be able to do that categorization. This can be considered for future studies.

Reviewer’s Comment: What is the difference between the Cude OR and Adjusted OR (AOR)? Please clarify the difference in the manuscript.

Authors Response: Crude odds ratio is used to show the difference in the change in an exposure variable influences the odds of the occurrence of an outcome variable while Adjusted odds ratio is used to show the difference in the change in an exposure variable influences the odds of the occurrence of an outcome variable, after controlling for other exposure variables in the model. 

Reviewer’s Comment: Please check the injury rate between males and females (Male 90 (66.7) and female 2 (66.7)).

Authors Response: We ha done re-analysis and presented the results on those injured and those who did not. 

Reviewer’s Comment: It is recommended that strategies or solutions which were presented at the end of each paragraph in the “Discussion” section, should be further clarified or explained in more detail, for instance, the acceptable level of lighting, workspace, or even other critical facilities influencing the occupational injuries.

Authors Response: The revised manuscript has these changes based on the recommendations by the reviewer. For instance, in the third paragraph of the discussion, some recommendations have been made to control occupational injuries occurrence. The revised section reads as;

We recommend that the management of the sawmill companies admonish the workers to be cautious by observing all protocols associated with their usage. Also, management can institute an award scheme for those who went through the year without injuries. 

Reviewer’s Comment: It is suggested that the study questionnaire be made available to the readers as a supplementary material.

Authors Response: The questionnaire has been attached as a supplementary file as suggested.

The training of the research assistants covered questionnaire (see Supplementary File 1)

Reviewer #2: Assessment Results

The study sought to assess the occupational injury rate and risk factors among workers among wood village in Ghana.

Reviewer’s Comment: The abstract was poorly formatted.

Authors Response: The abstract has been revised. 

Reviewer’s Comment: Results on the abstract was very scanty and a bit vague

Authors Response: We have revised the results section of the abstract as suggested please. It now reads as;

Results: Approximately 66.7% of the workers experienced occupational injuries within the 12 months preceding the study. Cuts (69.6%) were the commonly reported injuries. Injuries were mainly caused by machine parts/sharp objects (47.8%) and being hit by logs/objects (46.8%). Only 40.7% of the workers reported using PPE always and legs (38.0%) and hands (37.0%) were the common body parts injured. The worker’s monthly income, poor workspace design and poor lighting had increased odds of occupational injuries while an increase in age was associated with decreased odds of occupational injuries.

Reviewer’s Comment: Conclusion from the abstract was obvious based on anecdotal evidence and so nothing new was added

Authors Response: The conclusion of the abstract has been revised and it now reads as;

The prevalence of occupational injuries among the sawmill workers at the Sokoban Wood Village was high, and this calls for a need to prioritize promoting health and safety. Improving the safety of machine tools which accounted for almost half of the injuries, workspace design and lighting could be key for improving employee’s health and safety. 

Reviewer’s Comment: The objective of the study was not smart enough. Please recast the objective to make it smart.

Authors Response: This has been revised in the revised manuscript. It now reads as;

This study sought to assess the prevalence and factors associated with occupational injuries, including the possible effects of workspace design and lighting among small-scale sawmill workers at the Sokoban Wood Village, Kumasi, Ghana. 

Reviewer’s Comment: The authors claim they interviewed saw mill companies at Sokoban. However, a cursory look of Sokoban wood village can only reveal small holder saw millers and wood traders. This erroneous impression must be corrected.

Authors Response: We interviewed sawmill workers at the Sokoban Village as indicated in the manuscript. These are small-scale companies that are registered under the Furniture and Wood Workers Association of Ghana. As at the time the study was conducted there were only six small-scale companies with a total of 181 workers. These are not very big companies (like the sawmill companies located in the Asokwa/Ahisan industrial enclave in Kumasi. We mention this particular place because we suspect the reviewer most likely knows it and will better appreciate the comparison) but small-scale ones which are duly registered under the Furniture and Wood Workers Association of Ghana to operate. Clarity has been given in the revised manuscript. 

The Sokoban Wood Village was selected due to it being the biggest wood manufacturing site in Kumasi, Ghana with a total of six small-scale sawmill companies. These small-scale sawmill companies have been registered under the Furniture and Wood Workers Association of Ghana.

Reviewer’s Comment: The authors failed to give an account of the number of small holders saw millers traders they engaged with. Please state the number of registered saw miller shops visited.

Authors Response: There were six small-scale sawmill companies with a total of 181 sawmillers at the Sokoban Wood Village as at the time the study was conducted and we included all six companies in the study. This has been provided in the revised manuscript. 

The study population comprised sawmill workers from all six small-scale companies with a total of 181 staff at the Sokoban Wood Village. 

Reviewer’s Comment: Although the authors employed Cochrane’s formular in estimating the sample size, the sample size looked a bit on the low side, judging from nature of conclusion that the study intends to draw. The sample size used look inadequate.

Authors Response: This is true and it arises from the adjustment for a finite population and this reflect in the wide 95% confidence interval around some of the odds ratio estimates which we advised that such results should be interpreted with caution. The adjustment of the sample size with the finite population because the estimated sample size with the Cochran’s formular was higher than the total number of sawmill workers of 181 in the target population. This approach is well documented in the literature [1-4]. We have however, acknowledge the relatively small sample size as a study limitation in the revised manuscript. The study conclusion has also been revised to reflect to only our study participants. 

References 

1. Awini AB, Opoku DA, Ayisi-Boateng NK, Osarfo J, Sulemana A, Yankson IK, Osei-Ampofo M, Zackaria AN, Newton S. Prevalence and determinants of occupational injuries among emergency medical technicians in Northern Ghana. PLoS one. 2023 Apr 25;18(4):e0284943.

2. Bentum L, Brobbey LK, Adjei RO, Osei-Tutu P. Awareness of occupational hazards, and attitudes and practices towards the use of personal protective equipment among informal woodworkers: the case of the Sokoban Wood Village in Ghana. International Journal of Occupational Safety and Ergonomics. 2022 Jul 3;28(3):1690-8.

3. Mbaisi EM, Wanzala P, Omolo J. Prevalence and factors associated with percutaneous injuries and splash exposures among health-care workers in a provincial hospital, Kenya, 2010. Pan African Medical Journal. 2013 Apr 29;14(1).

4. Ofori AA, Osarfo J, Agbeno EK, Manu DO, Amoah E. Psychological impact of COVID-19 on health workers in Ghana: A multicentre, cross-sectional study. SAGE Open Medicine. 2021 Mar;9:20503121211000919.

Reviewer’s Comment: The authors did not state the type of cross-sectional study employed. But judging from the results presented, it seems to me the authors employed quantitative data collection approach. Generating frequency results from such a work cast doubts on the validity of the conclusion. It is therefore suggested that, the authors MUST collect data from another wood production village based on different criteria to server as control.

Authors Response: We adopted a quantitative study approach with an analytical cross-sectional design. We appreciate the reviewer’s comments, however the reason for calling for a new data collection is lost on us. The study is, essentially, exploratory and gives local insights for further studies. Furthermore, the work was done some years back and going back to collect new data is not practicable especially when time has evolved and we cannot compare what happened during that time to now. The dynamics may not be the same.

Reviewer’s Comment: For the quantitative study, the questions asked were a bit scanty. This makes the result a bit small to a made a very valid conclusion. It is suggested that more questions be added to expand the scope of questions.

Authors Response: We appreciate the reviewer’s comments, however, as stated earlier, there are practical constraints for us to undertake a new data collection exercise.

Reviewer’s Comment: It is also being suggested that some qualitative data should be taken from the study areas to add some contextual complexities to the quantitative results to be presented.

Authors Response: We appreciate the reviewer’s comments on taking some qualitative data from the study areas to add some contextual complexities to the quantitative results. However, we are unable to do so currently but will consider it for future studies.

Reviewer’s Comment: The authors claim they conducted multivariate analysis on some proportions of the respondents but the number is missing from the methodology. Please state the number of study subjects you conducted the multivariate analysis on.

Authors Response: We suspect the reviewer may have misunderstood us as no such claim was made. As is known, a regression analysis will run on complete datasets considering the variables involved. This will account for observed discrepancies in the study sample number and those reported in the regression. This is minimal and poses no problems for the results at all

Reviewer’s Comment: The authors employed only frequency assessment for Table 1 and 2. This is academically unacceptable for the kind of conclusions you intend to draw. Once a control group will be engaged, at least a Student T test analysis should be carried out to establish the appropriate level of significance in addition to the multivariate analysis.

Authors Response: This has been done as suggested by the reviewer. For instance, we used the student ttest to determine the average ages of workers who experienced injuries and those who did not (Table 1). 

Reviewer’s Comment: The authors claim they conducted multivariate analysis on both significant and non-significant results. Can you please explain the rationale behind this approach?

Authors Response: This was done because some variables in our study may not be significant but may be an important contributor to the occurrence of injuries at the workplace based on previous studies. These variables were included in the backward stepwise regression to control for possible confounding variables. This approach has been used in the literature (Nakua EK, Owusu-Dabo E, Newton S, Koranteng A, Otupiri E, Donkor P, Mock C. Injury rate and risk factors among small-scale gold miners in Ghana. BMC Public Health. 2019 Dec;19(1):1-8.)

The multivariable logistic regression analysis included all variables, both statistically significant (p ≤ 0.05) and non-significant variables in the univariate regression. The non-significant variables including age, work experience, cadre of work, work hours per week, provision of PPE and level of education were included in the model because they have been reported in the literature to have a significant effect on the occurrence of occupational injuries (17,18,31,32). A backward stepwise regression analysis (at p = 0.05) was used to control the non-significant variables that were deemed important. In the final multivariate model, an association was said to be statistically significant at a p-value <0.05.

Reviewer’s Comment: The authors must clearly state both inclusion and exclusion criteria during data collection

Authors Response: We state them in the original manuscript under ‘Study population.”. In the revised manuscript, we have stated them under a different heading named inclusion criteria, and exclusion criteria.

Inclusion criteria: Male and female sawmill workers aged 18 to 60 years who have been working with their respective companies for 1 year or more were included in the study.

Exclusion criteria: Workers who were not present on the days of the data collection were excluded from the study. 

Reviewer’s Comment: The study sought to assess the occupational injury rate and risk factors among workers among wood village in Ghana, however, data was collected from only a small area in Kumasi. This clearly cast doubt on the validity of your conclusion. On the basis of the above, I therefore reject the manuscript.

Authors Response: The title of the study has been revised to reflect where the study was conducted in Ghana. The revised title now reads;

Occupational injury prevalence and predictors among small-scale sawmill workers in the Sokoban Wood Village, Kumasi, Ghana.

Reviewer #3: 

Reviewer’s Comment: In the conclusion part of the abstract section, the presented conclusion is more general and can be presented without conducting research. The conclusion should be derived from the research results.

Authors’ Response: The conclusion of the abstract has been revised and it now reads as;

The prevalence of occupational injuries among the sawmill workers at the Sokoban Wood Village was high, and this calls for a need to prioritize promoting health and safety. Improving the safety of machine tools which accounted for almost half of the injuries, workspace design and lighting could be key for improving employee’s health and safety. 

Reviewer’s Comment: The present study lacks any particular innovation.

Authors’ Response: This has been addressed in the revised manuscript. The revised manuscript has accounted for an innovation in the study. This study reported on the possible effect of workspace design and lighting on the occurrence of occupational injuries which had previously not assessed in Ghana. 

This study, however, did not report on the effect the workspace design and lighting could have on the occurrence of occupational injuries among the sawmill workers. This study sought to assess the prevalence and factors associated with occupational injuries, including the possible effects of workspace design and lighting among small-scale sawmill workers at the Sokoban Wood Village, Kumasi, Ghana.

 Reviewer’s Comment: In Table 1, the number of married people is 41 and single people are 67, so their total is different from the total number of people under investigation. The sum of the mentioned percentages is also not correct.

Authors’ Response: This was an error and we have corrected it in the revised manuscript.

Reviewer’s Comment: Due to the existence of different types of PPE, it is not correct to mention the general title of PPE in this study. It should be specified what type of PPE is intended by the authors. Which PPE is provided by the employer and which is not. How much is the use of each PPE?

Authors’ Response: We agree with the reviewer on this comment. We have acknowledged this as a study limitation in the revised manuscript. The section in reference reads;

Again, this study did not assess the types of PPE which were provided to the workers….

Reviewer’s Comment: According to the results, about half of the workers have declared that safety and health instructions are available, and the other half have declared that the instructions are not available. Determining the level of access to procedures requires reviewing existing documents and cannot be achieved through questionnaires or interviews.

Authors’ Response: We agree with the reviewer on this comment. We have acknowledged this as a study limitation in the revised manuscript. The section in reference reads;

….. also did not review existing documentation on health and safety procedures. 

Reviewer’s Comment: According to the results, 89.9% have declared that the working environment is safe, while only 15.9% of the workers have declared that the design of the working environment is good. which indicates a contradiction in the results.

Authors Response: We have dropped the variable “working environment” in the revised manuscript and explanation has been given on how the variable workspace design was measured in the revised manuscript.

Reviewer’s Comment: Work environment design is a general term. It should be clear what the authors mean by the design of the work environment. What aspects of design are considered?

Authors Response: This has been provided in the revised manuscript as suggested. The section in reference now reads as;

Workspace design in this study referred to the space (shed) of the sawmill company and how the tools and logs have been arranged. Workspace design in this study was measured by asking the study participants if the tools and logs have been arranged without impeding accessibility (i.e. whether the space is confined or not) with a binary response of “good” or “poor”. A poor workspace design in this study meant that the space was confined (not spacious) or the arrangements of logs and work tools have created limited access for the workers while a good workspace design means that the workspace was not confined (spacious) or the workspace allowed easy accessibility.

Reviewer’s Comment: The numbers and percentages presented in Table 3 do not match each other.

Authors Response: This has been resolved in the revised manuscript.

Reviewer’s Comment: Contrary to the title, the main and technical risk factors have not been identified in this research.

Authors Response: The title of the study has been revised to prevalence and predictors. The title now reads as;

Occupational injury prevalence and predictors among small-scale sawmill workers in the Sokoban Wood Village, Kumasi, Ghana.

Reviewer #4:

Summary of the research and your overall impression

The research assesses the prevalence and risk factors associated with occupational injuries among sawmill workers in Ghana, specifically at one of the largest woodwork sites in the country, Sokoban Wood Village. The study aims to understand the extent of the problem and identify factors that contribute to these injuries.

The study concludes that:

1. Occupational injuries are a significant issue among sawmill workers in Ghana, with approximately two-thirds of the workers experiencing such injuries.

2. The most common mechanisms of occupational injuries in this population are being injured by machines or sharp objects and being hit by logs or objects.

3. Several risk factors are associated with occupational injuries, including the non-use of personal protective equipment (PPE), poor workspace design, and inadequate lighting.

4. The study found that increasing age and having a secondary education level are associated with a reduced likelihood of experiencing occupational injuries.

The conclusions drawn from this study are that the prevalence of occupational injuries among sawmill workers in Ghana is high, and interventions aimed at improving the health and safety of these workers should focus on younger employees and those with no formal education. Additionally, efforts should be made to enhance workspace design and lighting conditions in the workplace.

Strengths of the manuscript:

1. The study addresses an important occupational health issue in Ghana, shedding light on the prevalence and risk factors of occupational injuries among sawmill workers.

2. It uses a cross-sectional study design with a random sampling approach, which can provide valuable insights into the studied population.

3. The use of multivariate logistic regression allows for the identification of independent predictors of occupational injuries, contributing to a more comprehensive understanding of the problem.

Weaknesses of the manuscript:

Reviewer’s Comment: The sample size of 138 sawmill workers may be relatively small, and the findings may not be fully representative of all sawmill workers in Ghana.

Authors Response: The relatively small sample size has been acknowledged as a study limitation in the revised manuscript and have also recommended that the results be interpreted with caution

Some of the confidence intervals around the odds ratio estimates were wide due to the relatively small sample size, hence, we recommend that these results be interpreted cautiously

Reviewer’s Comment: The study relies on self-reported data, which may be subject to recall bias and may not capture all instances of occupational injuries accurately.

Authors Response: This was acknowledged in the original document as a study limitation. We have also given clarity on how the self-reporting could affect the study outcomes in the revised manuscript.

Despite these strengths, this study has some limitations including self-reporting of occupational injuries (not clinically confirmed) which can result in recall bias or misclassification of the outcome

Reviewer’s Comment: The manuscript lacks information on the specific interventions or recommendations for improving workplace safety beyond general suggestions.

Overall recommendation: The manuscript provides valuable insights into the prevalence and risk factors of occupational injuries among sawmill workers in Ghana. However, it would be beneficial to address the weaknesses mentioned above and provide more specific recommendations for interventions to improve workplace safety. Given the importance of the topic, I recommend accepting the manuscript with major revisions to address these concerns and enhance the overall impact of the study.

Authors Response: We are grateful to the reviewer for his insightful comments. We have revised the manuscript based on this and comments made by the other reviewers. The specific recommendations have been provided in the discussion of the revised manuscript. For instance, in the third paragraph of the discussion, some recommendations have been made to control occupational injuries occurrence. The revised section reads as;

We recommend that the management of the sawmill companies admonish the workers to be cautious by observing all protocols associated with their usage. Also, management can institute an award scheme for those who went through the year without injuries.

Major Issues

Reviewer’s Comment: Lack of Clear Differentiation from Existing Studies - The introduction does not sufficiently highlight how this study distinguishes itself from previous research, particularly the study titled "Characterizing occupational injuries among sawmill workers of Agbogbloshie timber market, Accra." While it mentions limited studies on occupational injuries among sawmill workers, it fails to specify what these studies did not address or what gaps they left unexplored. This omission makes it challenging for readers to understand the novelty or unique contribution of this study in relation to existing research. The authors should therefore highlight the novelty of the research.

Authors Response: The innovation in this study lies in its reportage on workspace design and lightning as factors that can affect injury occurrence in the sawmill industry regardless of the fact that we dealt with small-scale set-ups. The introduction of the study has been revised to reflect the gaps that our study will be filling. 

However, there are limited studies that assessed occupational injury rates as well as risk factors among sawmill workers in Ghana. A review of the literature showed that only one study reported on the factors contributing to occupational injuries among sawmill workers in Ghana but not in Kumasi (27). The study found that sawmill workers with no formal education had about two times increased risk of occupational injuries (27). In that study, the authors further reported injuries to the hands and legs (56%), shoulder and hand (12.7%) as the commonly reported occupational injuries among the sawmill workers (27). This study, however, did not report on the effect the workspace design and lighting could have on the occurrence of occupational injuries among the sawmill workers. This study sought to assess the prevalence and factors associated with occupational injuries, including the possible effects of workspace design and lighting among small-scale sawmill workers at the Sokoban Wood Village, Kumasi, Ghana. This study sought to bridge the literature gap by exploring the possible effect of workspace design, lighting and other factors on the occurrence of occupational injuries among small-scale sawmill workers.

Reviewer’s Comment: Unclear Significance of the Study - The introduction briefly mentions the prevalence of occupational injuries among sawmill workers in Ghana and the studies conducted among woodworking industry workers in Kumasi. However, it does not clearly establish why it is essential to address this issue or why it matters. The authors should elaborate on the significance of this research by explaining the potential impact of the findings on the health and safety of sawmill workers, workplace policies, and public health in Ghana. Providing a clearer rationale for the study would help readers better appreciate its importance.

Authors Response: This has been provided in the revised manuscript as suggested. 

This study, however, did not report on the effect the workspace design and lighting could have on the occurrence of occupational injuries among the sawmill workers. This study sought to assess the prevalence and factors associated with occupational injuries, including the possible effects of workspace design and lighting among small-scale sawmill workers at the Sokoban Wood Village, Kumasi, Ghana. This study sought to bridge the literature gap by exploring the possible effect of workspace design, lighting and other factors on the occurrence of occupational injuries among small-scale sawmill workers.

Reviewer’s Comment: The authors should clearly explain why they chose to use the Cochran's sample size formula instead of the Yamane (1967) formula: n=N/(1+Ne2). Cochran's sample size formula is for large-size populations and not usually used for the population of 181 people as indicated by the authors. Using the above formula would have resulted in a sample size of 125 instead of 110 and with a 25% non-response rate, the sample size will be 156 instead of 138. Clearly, the approach used in this study has reduced the sample size which affects the generalizability of the findings of the study. This should be stated as part of the limitations of the research.

Authors Response: The approach used to estimate the sample size in our study is supported by the literature [1-4]. We have however, acknowledge the relatively small sample size as a study limitation in the revised manuscript as recommended. 

Some of the confidence intervals around the odds ratio estimates were wide due to the relatively small sample size, hence, we recommend that these results be interpreted cautiously.

References 

1. Awini AB, Opoku DA, Ayisi-Boateng NK, Osarfo J, Sulemana A, Yankson IK, Osei-Ampofo M, Zackaria AN, Newton S. Prevalence and determinants of occupational injuries among emergency medical technicians in Northern Ghana. PLoS one. 2023 Apr 25;18(4):e0284943.

2. .

3. Bentum L, Brobbey LK, Adjei RO, Osei-Tutu P. Awareness of occupational hazards, and attitudes and practices towards the use of personal protective equipment among informal woodworkers: the case of the Sokoban Wood Village in Ghana. International Journal of Occupational Safety and Ergonomics. 2022 Jul 3;28(3):1690-8.

4. Mbaisi EM, Wanzala P, Omolo J. Prevalence and factors associated with percutaneous injuries and splash exposures among health-care workers in a provincial hospital, Kenya, 2010. Pan African Medical Journal. 2013 Apr 29;14(1).

5. Ofori AA, Osarfo J, Agbeno EK, Manu DO, Amoah E. Psychological impact of COVID-19 on health workers in Ghana: A multicentre, cross-sectional study. SAGE Open Medicine. 2021 Mar;9:20503121211000919.

Reviewer’s Comment: Backward Stepwise Approach with a p-value of 0.1: Using a p-value of 0.1 for variable selection in backward stepwise regression is less common and less stringent than the conventional p-value of 0.05. This approach may increase the risk of including variables that are not truly significant, potentially leading to overfitting and less robust results. It's advisable to use a more standard significance level of 0.05 for variable selection to ensure the reliability of the model. The authors should justify the selection of the p-value of 0.1 for variable selection.

Authors Response: A p-value of ≤ 0.05 have been used in the revised manuscript instead of the 0.1 which was queried by the reviewer. The revised section now reads;

The multivariable logistic regression analysis included all variables, both statistically significant (p ≤ 0.05) and non-significant including age, work experience, cadre of work, work hours per week, and level of education in the univariate regression. 

Reviewer’s Comment: Including Non-Significant Variables: The methodology mentions including both significant and non-significant variables from univariate analysis in the multivariable logistic regression. While it can be valuable to include theoretically important variables regardless of their statistical significance, it's crucial to justify why these non-significant variables are included. The rationale should be provided based on prior literature or theoretical considerations instead of only listing these sources.

Authors Response: We have listed the non-significant variables which were considered for the multivariate backward stepwise regression as well as providing justification for the reasons why these variables though not significant, were considered for selection into the model. 

Reviewer’s Comment: Clarify the Analysis Process: In the methodology section, the authors should explicitly clarify the sequence of analyses and their purposes. They should start by explaining that crude odds ratios were calculated at the univariate level to understand the individual associations between each independent variable and the outcome (occupational injuries). Then, describe the backward stepwise approach for variable selection in the multivariate model.

Authors’ response: This has been done in the revised manuscript as suggested by the reviewer. The revised section now reads as;

Univariate and multivariate logistic regression analyses were used to determine the factors associated with occupational injuries among the sawmill workers. Crude odds ratios were estimated at the univariate level to understand the individual associations between each predictor and the outcome (occupational injuries) variables. The variable ‘sex’ of the study participants was excluded from the univariate analysis due to the fact that it was essentially homogenous. The multivariable logistic regression analysis included all variables, both statistically significant (p ≤ 0.05) and non-significant variables in the univariate regression. The non-significant variables including age, work experience, cadre of work, work hours per week, provision of PPE and level of education were included in the model because they have been reported in the literature to have a significant effect on the occurrence of occupational injuries (17,18,31,32). A backward stepwise regression analysis (at p = 0.05) was used to control the non-significant variables that were deemed important. In the final multivariate model, an association was said to be statistically significant at a p-value <0.05.

Reviewer’s Comment: Specify the Purpose of Crude Odds Ratios: The authors should clearly state that the purpose for the calculation of crude odds ratios at the univariate level. 

Authors’ response: This has been done in the revised manuscript as suggested by the reviewer. The revised section now reads as; Crude odds ratios were estimated at the univariate level to understand the individual associations between each predictor and the outcome (occupational injuries) variables.

Reviewer’s Comment: Multivariate Model Development: The authors should describe the development of the multivariate logistic regression model using the backward stepwise approach.

Authors’ response: This has been done in the revised manuscript as suggested. It now reads as;

The multivariable logistic regression analysis included all variables, both statistically significant (p ≤ 0.05) and non-significant variables in the univariate regression. The non-significant variables including age, work experience, cadre of work, work hours per week, provision of PPE and level of education were included in the model because they have been reported in the literature to have a significant effect on the occurrence of occupational injuries (17,18,31,32). A backward stepwise regression analysis (at p = 0.05) was used to control the non-significant variables that were deemed important. In the final multivariate model, an association was said to be statistically significant at a p-value <0.05.

Reviewer’s Comment: Handling Categorical Data: The authors should explain how categorical data were coded for inclusion in the model and provide details on any reclassifications or transformations performed on the categorical variables. They should also clarify the reference categories for each categorical variable used in the analysis.

Authors’ response: This information has been provided in the revised manuscript. There was no reclassification or transformation of any variable in this study. The reference categories have also been provided in the revised manuscript 

Reviewer’s Comment: Handling Missing Data: The authors should describe how missing data were handled in the analysis. Specify whether any imputation methods were used, or if data were excluded, provide reasons for data exclusions. Transparency in handling missing data is crucial for the validity of the results.

Authors’ response: There were no missing data in the study results. 

Reviewer’s Comment: Testing for Collinearity: The authors should explain how they tested for multicollinearity among the independent variables prior to fitting the multivariate model. They should describe the methods used to assess collinearity (e.g., variance inflation factor or correlation matrices) and report the results. If collinearity was detected, they should explain how it was addressed.

Authors’ response: This has been provided in the revised manuscript as suggested by the reviewer. The revised section now reads;

The Variance Inflation Factor (VIF) was used to assess the multi-collinearity of key predictor variables. There was no evidence of multi-collinearity between the key predictor variables from the VIF results. (Mean VIF = 1.19, Maximum VIF = 1.36, Minimum VIF = 1.10) (see S1 Table). The fit of the model was assessed using the Hosmer-Lemeshow test and the results did not detect evidence of poor fit (P = 0.190).

Reviewer’s Comment: Presentation of p-Values: The authors should present p-values for each variable in the final multivariate model. This is essential for assessing the statistical significance of each predictor in relation to the outcome.

Authors’ response: This has been provided as recommended by the reviewer.

Reviewer’s Comment: The conclusion of this research is a repetitive summary of the findings already presented. To improve on this, the following are suggested for the authors: Restate the objective and indicate whether it was answered or achieved.

Authors’ response: The conclusion of the study has been revised to avoid the repetition of results as suggested by the reviewer. The revised section now reads as;

Our study revealed that the prevalence of occupational injuries among sawmill workers in the Sokoban Wood Village, Kumasi was high which raises concerns for public health. Worker’s age, monthly income, workspace design and lighting at the workplace were the determining factors for the occurrence of occupational injuries. The commonly reported mechanism of occupational injuries was injury by machine parts/sharp objects and being hit by logs/objects. Factors such as the age of workers, monthly income, workspace design and lighting at the workplace were associated with occupational injuries. We recommend that the management of the sawmill companies should ensure that the arrangements of logs and working tools at the sawmills do not affect accessibility and that lighting at the workplace provides adequate visibility which will not result in workers straining their eyes or working in an awkward posture. It is also recommended that the management of the sawmill companies should ensure regular maintenance of work machines and enforce increased use of PPE by the workers. Future studies can evaluate the impact of regulatory or legislative agencies on the employees’ health and safety in the small-scale sawmill industry. This study provides useful information to inform policies to champion employees’ health and safety in small-scale sawmill companies. 

Reviewer’s Comment: Discuss the implications of the findings and explain why they matter, including any practical applications or theoretical implications.

Authors’ response: This has been done as suggested by the reviewer. 

Reviewer’s Comment: Acknowledge any limitations or weaknesses of the study and suggest directions for future research, including any areas where further investigation is needed.

Authors’ response: This has been done as suggested. 

The study has some limitations including self-reporting of occupational injuries (not clinically confirmed) which can result in recall bias or misclassification of the outcome. However, to avoid misclassification of occupational injury, study participants were asked to detail how and where the injury occurred. The assessment of occupational injuries in the last twelve months preceding the data collection can result in an underestimation of the prevalence rate due to recall bias (participants had to recall the event in the last 12 months). Some of the confidence intervals around the odds ratio estimates were wide due to the relatively small sample size, hence, we recommend that these results be interpreted cautiously. Again, this study did not assess the types of PPE which were provided to the workers and also did not review existing documentation on health and safety procedures. Also, workspace design and lighting were subjectively assessed. Nevertheless, the findings of this study are insightful for further more comprehensive investigations in the future. 

Future studies can evaluate the impact of regulatory or legislative agencies on the employees’ health and safety in the small-scale sawmill industry.

Reviewer’s Comment: Missing references

Authors’ response: This has been done as suggested please.

Reviewer’s Comment: The last section of this part of the manuscript should be supported by a reference or removed “employment to the majority of the youth in Kumasi, Ghana”. The assertion that the Sokoban wood village provides employment to the majority of youth in Kumasi should be verified since it could be misleading.

Authors’ response: This part has been removed from the manuscript at suggested. 

Reviewer’s Comment: The authors should define the roles of the sawmill workers involved in the study. For instance, what do the authors mean by Engineers, Operators and Technicians? This is not clear and could be used to refer to the same type of workers.

Authors’ response: The variable has been recategorized as cadre of staff with categories such as administrative and non-administrative staff in the revised manuscript. We have explained in the footnote of Table 1 the staff which constitutes both categories.

Reviewer’s Comment: The authors should provide a justification for excluding those who had worked for less than a year? Due to the inexperience of new hires, they are exposed to more work injuries compared to experienced ones. This has been well established in existing literature.

Authors’ response: Those who had worked for less than a year was excluded from this study because we estimated injuries in the preceding 12 months to the data collection. Clarity has been given in the revised manuscript. 

The outcome variable of this study was a self-reported injury associated with any sawmilling activities in the twelve (12) months preceding the data collection.

Reviewer’s Comment: How was the study population obtained? The authors should indicate the source of that information.

Authors’ response: This was provided in the original manuscript and clarity has also been given to the study population and the source in the revised manuscript as recommended. The revised section now reads;

The study population comprised all sawmill workers from all six sawmill companies at the Sokoban Wood Village.

Reviewer’s Comment: The authors should indicate how many sawmill companies were available at the study site and how they selected the six sawmill companies for the study. Do the six companies represent only, say, 10% of the total number of companies at Sokoban? This should be made clear in the manuscript so the applicability of the findings from the research can be identified by readers.

Authors’ response: There were six sawmill companies at the Sokoban Wood Village and we included all of them in the study. Clarity has been given in the revised manuscript. 

A total of 138 sawmill workers were selected through a simple random sampling technique from all the existing six sawmill companies at the Sokoban Wood Village

Reviewer’s Comment: How was the probability proportion approach used? How many respondents were selected from each company, and how many were there in total?

Authors’ response: We have clarified how the proportional allocation was done in the revised manuscript.

The proportional allocation of the study participants was derived by dividing the number of sawmill workers in each company by the total number of workers (181) in all the six companies and multiplied by the estimated sample size of 138.

Reviewer’s comment: Table 2: The variables used for each of these parameters should be explained. Workplace environment (Safe, Not safe) What do the authors mean by safe workplace environment and vice versa?

Authors’ response: We have explained all the variables in Table 2 under subsection “predictor variables” However, we have dropped the variable “workplace environment” from the manuscript 

Reviewer’s comment: Workspace design (Good or poor) What do the authors mean by good workspace design and vice versa?

Authors’ response: This has been explained in the revised manuscript under subsection “predictor variables”. The revised section now reads as;

Workspace design in this study referred to the space (shed) of the sawmill company and how the tools and logs have been arranged. Quality of workspace design was measured by asking the study participants if the tools and logs have been arranged without impeding accessibility (i.e. whether the space is confined or not) with a binary response of “good” or “poor”. A poor workspace design in this study meant that the space was confined (not spacious) or the arrangements of logs and work tools have created limited access for the workers while a good workspace design means that the workspace was not confined (spacious) or the workspace allowed easy accessibility.

Reviewer’s comment: Lighting at work (Good or poor) What do the authors consider as good lighting at work and vice versa?

Authors’ response: This has been explained in the revised manuscript under subsection “predictor variables”. The revised section now reads as;

Lighting at the workplace in this study was measured by asking the study participants whether there was sufficient lighting irrespective of the source if they could accomplish tasks freely without having to either strain their eyes for vision or work in an awkward posture, with a binary response of ‘good’ or ‘poor’. ‘Good lighting’ in this study meant that there was sufficient lighting at the workplace which allowed workers to accomplish tasks freely without having to either strain their eyes for vision or work in an awkward posture while ‘poor lighting’ in this study meant that there was insufficient lighting at the workplace which either affected workers vision or working posture. 

Reviewer’s comment: Data presentation: The authors should provide the questionnaire used for data collection as a supplementary file to the manuscript.

Authors’ response: This has been provided in the revised manuscript as suggested by the reviewer.

Reviewer’s comment: Regarding the assessment of the availability of health and safety procedures, it may be more efficient to determine whether these procedures exist at the company level rather than asking each individual worker. It could be more informative if the authors report the presence or absence of these procedures for each of the six companies separately. This would clarify how many out of the six companies have established health and safety procedures and how many do not. Alternatively, if this information cannot be obtained accurately, it might be advisable to consider excluding this aspect from the results to prevent potential confusion in its current format.

Authors’ response: We are unable to provide the information on the existence of the health and safety procedures existence in all six companies. Hence, we have dropped that variable in the revised manuscript as suggested by the reviewer.

---

## [Decision Letter · Decision Letter 1]

24 Jan 2024

PONE-D-23-20985R1Occupational injury prevalence and predictors among small-scale sawmill workers in the Sokoban Wood Village, Kumasi, GhanaPLOS ONE

Dear Dr. Opoku,

Thank you for submitting your manuscript to PLOS ONE. After careful consideration, we feel that it has merit but does not fully meet PLOS ONE’s publication criteria as it currently stands. Therefore, we invite you to submit a revised version of the manuscript that addresses the points raised during the review process.

Authors have been able to improve upon the manuscript significantly based on the reviewers' comments shared earlier on. This is a very positive progress on the article. Meanwhile, there are few minor comments on the methodology that require authors attention for the manuscript to meet the standard of our cherished journal. the details are provided in the comments to authors and briefly share here as well:- A reference citation needed at the first use of Cochrane formulae- sampling size narration can be made clearer. - validity and reliability of the data collection tools still outstanding and unclear and needs to be assessed to give confidence to data, analysis and policy decision coming out of the work.==============================

We look forward to receiving your revised manuscript.

Kind regards,

Prof. Bismark Dwumfour-Asare, MSc, PhD

Academic Editor

PLOS ONE

Journal Requirements:

Additional Editor Comments:

Minor comment on the Methodology

- Cochrane formular used must be referenced/source cited, first formular written must be given the equation number and reference/citation.

- It’s not clear why adjusting for the 25% (quite high) non-response rate and go for the 138? Meanwhile, you targeted all staff 181 (census sample) and that should be enough to indicate and then report on the response/non-response rate.

- How was the internal validity and reliability done? Was there any computation for the Cronbach alpha values etc? it’s not enough to say use of experienced staff/members improved validity and reliability – where is the evidence? this validity and reliability section is key and how that is ensured need scientific basis (universal approach) like Cronbach alpha computation/analysis.

Reviewers' comments:

Reviewer's Responses to Questions

**Comments to the Author**

1. If the authors have adequately addressed your comments raised in a previous round of review and you feel that this manuscript is now acceptable for publication, you may indicate that here to bypass the “Comments to the Author” section, enter your conflict of interest statement in the “Confidential to Editor” section, and submit your "Accept" recommendation.

Reviewer #1: All comments have been addressed

2. Is the manuscript technically sound, and do the data support the conclusions?

Reviewer #1: Yes

3. Has the statistical analysis been performed appropriately and rigorously? 

Reviewer #1: (No Response)

4. Have the authors made all data underlying the findings in their manuscript fully available?

Reviewer #1: Yes

5. Is the manuscript presented in an intelligible fashion and written in standard English?

Reviewer #1: Yes

6. Review Comments to the Author

Reviewer #1: The authors mainly applied the reviewers’ comments. It could benefit sawmill workers' knowledge and perception of OHS principles.

7. PLOS authors have the option to publish the peer review history of their article (what does this mean?). If published, this will include your full peer review and any attached files.

Reviewer #1: No

---

## [Author Response · Author response to Decision Letter 1]

27 Jan 2024

Reviewer’s comment: Cochrane formula used must be referenced/source cited, first formula written must be given the equation number and reference/citation.

Authors’ response: This has been done as suggested. 

The Cochrane formula was used to estimate the sample size for this study [28]:

Sample size,SS= (Z^2 pq)/d^2 , Equation 1 [28]

Reviewer’s comment: It’s not clear why adjusting for the 25% (quite high) non-response rate and go for the 138? Meanwhile, you targeted all staff 181 (census sample) and that should be enough to indicate and then report on the response/non-response rate.

Authors’ response: Please, we did not target all staff. The study population was rather limited to small-scale sawmill workers in the Sokoban Wood Village. We have revised the statement in the methods which seeks to portray that we targeted all staff of 181 to provide more clarity. We have also provided an explanation on the adoption of the 25% non-response rate in the revised manuscript. The section in reference now reads as;

Study population

The study population was limited to small-scale sawmill workers at the Sokoban Wood Village in Kumasi.

Sample size estimation and sampling procedure 

The Cochrane formula was used to estimate the sample size for this study [28]:

Sample size,SS= (Z^2 pq)/d^2 , Equation 1 [28]

where Z is the reliability coefficient of 1.96 at a 95% confidence interval, p is the proportion of sawmill workers that sustained occupational injuries, q is (1 – p) and ‘d’ is the degree of accuracy. Assuming a 78.0% prevalence of occupational injuries among sawmill workers from an earlier study [11], and a 5% degree of accuracy, a sample size of 264 was estimated. 

Sample size,S= (〖1.96〗^2 x 0.78 x (1-0.78))/〖0.05〗^2 =264 Equation 1

We adjusted the sample size to 110 for a finite population, N, of 181 (which is the total number of small-scale sawmill workers at the Sokoban Wood Village) using Equation 2. This approach is well documented in the literature [23,29–31]. The 181 is the total number of all workers from the small-scale companies. Knowing this was a finite population, we applied the adjustment for it and got the initial sample size.

Adjusted sample size,Ass= S/(1+ (S-1)/N ) Equation 2

where Ass is the new sample size, S is the estimated sample size, 264, and N is the finite population. Adjusting by 25% for non-response, a total of 138 sawmill workers were recruited for the study. We used 25% because initial reconnaissance assessments showed it was prudent to anticipate that interviews could span over 2 days because of the volume of work for the workers. On this basis, there was no guarantee a worker who had commenced the interview the previous day would come to work the next day. To overcome these uncertainties, we used the relatively high 25% adjustment factor.

Reviewer’s comment: How was the internal validity and reliability done? Was there any computation for the Cronbach alpha values etc? it’s not enough to say use of experienced staff/members improved validity and reliability – where is the evidence? this validity and reliability section is key and how that is ensured need scientific basis (universal approach) like Cronbach alpha computation/analysis.

Authors’ response: The section in reference has been revised to reflect that a pretest was conducted to improve the reliability of the questionnaire. Again, the assessment of occupational injuries and other variables of interest in this study were not assessed with Likert scale responses. Hence, we are unable to use Cronbach’s alpha computation/analysis. We have also acknowledged the questionnaire (tool) as a study limitation because it was not validated and recommended readers to be cautious in the interpretation of the results. 

The questionnaire was not validated; however, it was pre-tested among fifteen (15) small-scale sawmill workers in a different district that was not part of the source population to check to improve its reliability and validity. All the necessary corrections were effected before using it for the final data collection

Even though the questionnaire was pretested and fine-tuned following that, the lack of a comprehensive evaluation of validity and reliability is a limitation and the findings ought to be taken with caution.

---

## [Editor Report · Decision Letter 2]

2 Feb 2024

Occupational injury prevalence and predictors among small-scale sawmill workers in the Sokoban Wood Village, Kumasi, Ghana

PONE-D-23-20985R2

Dear Dr. Opoku,

We’re pleased to inform you that your manuscript has been judged scientifically suitable for publication and will be formally accepted for publication once it meets all outstanding technical requirements.

Kind regards,

Prof. Bismark Dwumfour-Asare, MSc, PhD

Academic Editor

PLOS ONE

Additional Editor Comments (optional):

Congratulations for successfully responding to all comments raised to improve your paper. Please, kindly capture all the limitations articulated in your responses t at the "Strengths and limitations of the study" sub section.
---

## [Editor Report · Acceptance letter]

28 Mar 2024

PONE-D-23-20985R2 

PLOS ONE

Dear Dr. Opoku, 

I'm pleased to inform you that your manuscript has been deemed suitable for publication in PLOS ONE. Congratulations! Your manuscript is now being handed over to our production team.

Kind regards, 

on behalf of

Prof. Bismark Dwumfour-Asare 

Academic Editor

PLOS ONE